# Mesosphere of Carbon-Shelled Copper Nanoparticles with High Conductivity and Thermal Stability via Direct Carbonization of Polymer Soft Templates

**DOI:** 10.3390/ma15217536

**Published:** 2022-10-27

**Authors:** Min Huang, Xinyu Cao, Jingnan Zhang, Huiqun Liu, Jiaxin Lu, Danqing Yi, Yongmei Ma

**Affiliations:** 1School of Material Science and Engineering, Central South University, Changsha 410083, China; 2Institute of Chemistry, Chinese Academy of Sciences, Beijing 100190, China

**Keywords:** anti-oxidation, electrical properties, nanocomposites

## Abstract

Copper nanoparticle (Cu NP) is a promising replacement for noble metal nanoparticles due to its high electrical conductivity and low cost. However, Cu NPs are relatively active compared to noble metals, and current ways of protecting Cu NPs from oxidation by encapsulation have severe drawbacks, such as a long reaction time and complicated processes. Here, a facial and effective method to prepare the mesosphere of carbon-shelled copper nanoparticles (Cu@MC) was demonstrated, and the resulting Cu@MC was both highly electrically conductive and thermally stable. Cu@organic (100 nm) was first synthesized by the reduction of Cu ions with poly (vinyl pyrrolidone) (PVP) and sodium poly ((naphthalene-formaldehyde) sulfonate) (Na-PNFS) as soft templates. Then, the carbon shells were obtained by in situ carbonization of the polymer soft templates. The Cu@organic and Cu@MC showed an anti-oxidation ability up to 175 and 250 °C in the air atmosphere, respectively. Furthermore, the Cu@MC exhibited excellent volume resistivity of 7.2 × 10^−3^ Ω·cm under 20 MPa, and showed promising application potential in electric sensors and devices.

## 1. Introduction

Copper nanoparticle (Cu NP) has been considered as a promising replacement for the expensive noble metal nanoparticles. Copper not only exhibits high electrical conductivity, which is only 6% lower than silver, but also dramatically lowers the cost due to its abundance on Earth. Furthermore, as cooper can be flexibly processed, Cu NPs show broad application potential in electronic and optical devices, etc. However, copper is relatively unstable compared to noble metal. When exposed to air, copper nanoparticles can be oxidized much more quickly than noble metals, which severely reduces the electrical conductivity [1]. 

Encapsulation of Cu NPs is an effective way to protect them from oxidation. Methods of protecting Cu NPs by shell and by noble metals [2,3,4,5], organic layers [6,7,8], and carbon or graphene layers [9,10] have been reported. The protective layers can protect the metal core from oxidation, as well as maintaining the high conductivity of Cu NPs. It has been reported that for some other metal nanoparticle-embedded meso-structures, carbon layers or frameworks can be achieved by pyrolysis of carbon precursors [11,12,13]. Several groups [14,15,16] have reported about capping the Cu NPs with a few layers of graphene, either by flame spray synthesis or metal–organic chemical vapor deposition. The graphene-encapsulated Cu NPs stay stable up to 165 °C in the air, and the electrical conductivity of the Cu NPs without sintering was ~1.56 S cm^−1^. Cu@carbon NPs, produced by the chemical vapor deposition of multilayer graphene, were reported with an electrical resistivity of 1.7 × 10^−4^ Ω·cm [17,18]. However, most of these methods require complicated processes, long reaction times, and gas carbon sources. 

In this paper, we report a new, simple, and effective method to prepare the mesosphere of carbon-shelled copper nanoparticles (Cu@MC) by using the direct carbonization of the soft template of Cu NPs. First, sodium poly ((naphthalene-formaldehyde) sulfonate) (Na-PNFS) and poly (vinyl pyrrolidone) (PVP) were used as soft templates to synthesize organic-capped copper nanoparticles, and then carbon shells around the nanoparticle could be formed in situ by treatment under 900 °C. This preparing method of encapsulation of Cu NPs is time- and energy-saving. The mechanism of the Cu@MC formation and the stability and conductivity of the produced Cu NPs were analyzed.

## 2. Materials and Methods

### 2.1. Preparation of Copper Nanoparticles Capped with Organic Layers (Cu@organic)

Cu@organic was synthesized by reducing CuSO_4_·5H_2_O with hydrazine in a solution. Na-PNFS (99%, Mn = 2000) and poly (vinyl pyrrolidone) (PVP, 99%, K-30) were used as the capping agents. Na-PNFS (5 g/L) and PVP (9 g/L) were dissolved in water, to which CuSO_4_ (10 mM) was added under stirring. Then the pH value was adjusted to 10 ± 0.2 by saturating the NaOH solution, and the mixture was kept in a thermostatic water bath at 50 °C for 20 min. Finally, N_2_H_4_·H_2_O (n [N_2_H_4_]: n [CuSO_4_] = 66:1) was added. The thermostatic process was continued for another 90 min. The final reaction mixture was a uniform stable colloidal dispersion. The Cu NPs were washed with water and collected by centrifugation until the upper liquid was colorless.

### 2.2. Preparation of Mesosphere of Carbon-Shelled Copper Nanoparticle (Cu@MC)

The collected Cu NPs were loaded into a furnace chamber and heated to 900 °C for 2 h under 50 sccm of argon gas supply. The ramping rate of the temperature was approximately 5 °C/min followed by natural cooling to room temperature with an identical gas configuration as the annealing condition.

### 2.3. Preparation of Copper Nanoparticle-Based Conductive Paste (Cu Paste)

The paste consists of Cu@MC, matrix polymer, ethyl cellulose (EC), and propylene- glycol monomethyl ether (PGME). The base paste was obtained by dissolving EC in PGME at 80 °C and stirred for 12 h with a mass ratio of 10%. Then, the Cu paste was prepared by mixing Cu@MC with the base paste in a mass ratio of 2:1.

### 2.4. Characterization

Particle size and distribution were characterized by dynamic light scattering (DLS, Malvern) with water as a dispersant. Transmission electron microscope (TEM) images and selected area electron diffraction (SAED) patterns were obtained by a TEM (JEOL-2200FS) operated at 200 kV. X-ray diffraction (XRD) spectra were collected on an Empyrean-2 X-ray diffractometer (PANalytical, Cu Kα). The surface chemistry of the organic-capper copper nanoparticles and Cu@MC was characterized using an XPS (ESCALAB250XI, K-alpha) system equipped with a monochromatic Al Kα source (hν = 1486.6 eV). Raman spectra were recorded on a Raman spectrometer (LabRAMHR). To evaluate the anti-oxidation properties of coated copper particles, thermogravimetric analysis (TGA) was conducted on a Perkin–Elmer Instruments Pyris-1 at a ramping rate of 20 °C/min under air atmospheres. Thermogravimetric analysis (TGA) was conducted on a Perkin–Elmer Instruments TGA 8000 at a ramping rate of 20 °C/min under nitrogen atmospheres to characterize the carbonization process of PVP and Na-PNFS. The volume resistivity was measured by a powder resistivity tester (FT-300I, Ningbo Rooko Instrument Co., Ltd., Ningbo, China) under 20 MPa. The surface resistivity of copper paste was obtained by a four-point probe (RTS-8, Guangzhou Four Probe Tech., Guangzhou, China) measurement of the paste film, which was printed on a glass substrate and dried under ambient conditions for 3–4 h. 

## 3. Results

The reaction process and the proposed mechanism of forming organic polymer-wrapped copper nanoparticles (Cu@organic) and Cu@MC are illustrated in Figure 1 and Figure 2. The color changes of the reaction solution indicated formations of different organometallic complex particles. A detailed description of the experimental phenomena is presented in the Appendix A. It suggests that Na-PNFS and PVP can form polymer–surfactant complexes just like PVP and SDS [19], and a uniform stable colloidal dispersion was obtained as the product (d). The XRD, DLS, and TEM characterization (Figure 3, Figure 4 and Figure 5) further verify the formation of reddish-brown Cu@organic (e) and dark brown Cu@MC (f).

The XRD patterns (Figure 3) of the as-prepared products (e) and (f) were almost identical and only showed three main characteristic peaks at 43.3°, 50.5°, and 74.1°, corresponding to the three crystalline planes of (111), (200), and (220) of the face-centered cubic (fcc) Cu, respectively. No diffraction peaks of copper oxide were present. This indicates that the reduction of Cu^2+^ was successful, and the copper particles were stable during carbonization.

The particle size and distribution were characterized by a dynamic light scattering (DLS) measurement and TEM observation. The DLS results (Figure 4) suggest that in the reactions of a–e in Figure 1, copper nanoparticles with a size below ~100 nm were obtained. TEM observations in Figure 5a–c also show that product (e) was nanoparticles with a size of around 100 nm wrapped with shell materials. The selected area electron diffraction (SAED) pattern obtained from the black grains of product (e) (insertion in Figure 5c) show four fringe patterns with plane distances of 2.09, 1.81, 1.28, and 1.09 Å that are indexed to the (111), (200), (220), and (311) planes of face-centered cubic (fcc) Cu, respectively. It verified that the nano-particle core is copper. The shell material of the product € nanoparticle should be the soft templates consisting of Na-PNFS and PVP, and it was further verified by FTIR, as shown in Figure 6. Under a high-magnification TEM micrograph (Appendix A), the shape of the nanoparticle was not perfectly spherical. The edges may relate to the absorption of surfactant which can control the particle morphology during the reduction process. 

After treatment at 900 °C for 2 h in the argon atmosphere, a TEM image of product (f) shows that the particle size increased and the main particle size distribution lay in the range of 100~200 nm (Figure 5d–f). This indicates that the particle size grew by the smaller particles’ agglomeration or aggregated into bigger ones. Firstly, during annealing, the organic layer consisted of polymer–surfactant complexes that prevented particles aggregate from being decomposed. Secondly, the melting temperature can be much lower for Cu NPs, especially at the surface. It can also cause the agglomeration of the particles [20]. The shell material of product (f) was further confirmed as carbon layers by Raman spectrum. Under a high-magnified TEM observation, it was found that the thickness of the carbon layers was not uniform and had various thicknesses. The boundary of the copper core was fuzzy, and the edge was not clear-cut with the carbon layer. 

The enlarged TEM image of Cu@organic (Figure 5a–c) shows that around the dark nanoparticle, which is copper, a gray layer could be observed which should be the soft templates consisting of Na-PNFS and PVP, and it was further verified by FTIR, as shown in Figure 6. The FTIR of Cu@organic shows a spectrum with the combined characteristics of those of Na-PNFS and PVP. The two peaks at 1640 and 1580 cm^−1^ were from the vibrations of C=O in the five-lactam ring of PVP and naphthalene ring skeletons, respectively. The bands at 1194 and 1116 cm^−1^ represent the vibrations of the S=O groups. Compared with the peaks of the S=O groups in Na-PNFS (1189 and 1121 cm^−1^), the peaks shifted. This indicates that there was a chemical interaction between the Cu core and Na-PNFS. These peaks verified that the product “e” was copper nanoparticles wrapped with PNFS and PVP that were labeled as Cu @organic NPs. In the spectrum of Cu@MC, the peaks assigned to the S=O groups (1194 cm^−1^) and C=O groups (1116 cm^−1^), which derive from organic surfactants, disappeared. A broad new peak appeared at 1024 cm^−1^, and this may relate to the C-S, C-N hybridized structure. Combined with the Raman spectrum, it can be confirmed that the organic shell layer converted into a carbon shell.

The XPS results agree that the shell layer of the Cu@organic is composed of PVP and Na-PNFS. In the XPS spectrum of Cu 2p3/2 (Figure 7a), the peak could be deconvoluted into two peaks: 932.4 and 934.4 eV, which were assigned to the Cu and Cu-organic complex. For PVP and Na-PNFS, if there was no interaction with copper, the bonding energy of the O1s signal in C=O and S=O would appear at 529.7 [19] and 531.9 eV (Figure 7c), respectively. The O1s signal of the organic-capped copper nanoparticles can be deconvoluted into three peaks: 530.7, 531.9, and 533.3 eV (Figure 7b). These peaks are from the carboxyl oxygen atoms (C=O) of the PVP repeating units that are interacting with the Cu core, the S=O of the poly ((naphthalene-formaldehyde) sulfonate) that is not interacting with the Cu core, and the S=O of the poly ((naphthalene-formaldehyde) sulfonate) that is interacting with the Cu core, respectively. Both PVP and Na-PNFS had a lone pair electron, which could interact with Cu^2+^ and create an electrostatically bond to the copper. This indicates that the organic layers were chemisorbed to the Cu surface via a chemical interaction. 

After sintering, the copper particles in product “f” were still wrapped by a layer (Figure 5d–f). Both dense and fuzzy layers can be observed [21]. The Raman analysis of product “f” (Figure 8) confirmed the formation of carbon layers on the outer shells of copper nanoparticles. The first-order region of Raman spectra showed position D and G peaks with high intensities near 1345 and 1600 cm^−1^, respectively, and a broad peak was observed in the range of 2300−3200 cm^−1^. These are the character peaks of carbon material. The D band arose from defects in the hexagonal sp^2^ carbon network or the finite particle-size effect [22,23], while the G band originated from the stretching motion of the sp^2^ carbon pairs in both rings and chains [22]. The intensity ratio of the D and G bands (I_D_/I_G_) was determined to be 0.78, implying a certain degree of graphitization. These results indicate that both amorphous C and the nonplanar structure of graphene occurred during the annealing process. The growth mechanism of carbon layers on Cu NPs can be explained as follows. Firstly, the organic shell was converted to a-C (amorphous carbon) layers through a pyrolysis process. Then, the a-C layers in direct contact with the Cu nanoparticles were converted to graphitic ordering due to the catalytic effect of Cu [24]. Thus, the copper nanoparticles might have a graphene-like inner shell, and an amorphous outer carbon shell. Thus, the product “f” consisted of copper nanoparticles encapsulated in the carbon-shelled mesosphere, and “f” was labeled as Cu@MC. 

The critical micelle concentration of the Na-PNFS was 1.69 mmol/L [25]. Therefore, when the Na-PNFS was added at a concentration of 5 g/L (or 2.5 mmol/L), as in the experiment, micelles were formed in the solution. With the presence of PVP, the micelles could interact with PVP to form polymer–surfactant complexes [26]. The copper nanocrystalline was formed inside the micelles, and then clustered as larger particles with the help of complexes, and eventually transformed into carbonated shells after annealing [27,28].

Thermogravimetric analysis (TGA) tests of Na-PNFS and PVP show that Na-PNFS have a 60% weight remaining at 800 °C PVP, while PVP is almost completely decomposed with only 2.3% remaining at 750 °C (Figure 9a). The Raman spectra of Na-PNFS and PVP annealing at 900 °C for 2 h in a nitrogen atmosphere show that the residues were graphitized materials (Figure 9b). This result indicates that the carbon layer on the copper surface was mainly derived from Na-PNFS, which was the soft template when Cu NPs were formed. 

The oxidation of copper nanoparticles can be observed through the weight gain of the sample during oxidation [29]. TGA shows that the Cu@organic were stable until 175 °C, as no weight gain occurred before this temperature, while the Cu@MC were stable up to 250 °C. The weight gain of Cu@organic at 550 °C was 20%, and Cu@MC at 550 °C was 10% (Figure 10). The increased stability suggests that the carbon layer provides good protection for the copper nanoparticles from oxidation. 

The Cu@MC pellet had a high electrical resistivity of 7.27 × 10^−3^ Ω·cm at room temperature (Table 1), which is about 5 times higher than the graphene-wrapped Cu NPs reported previously. The high stability and conductivity suggest that the copper nanoparticles were fully wrapped by the carbon shell. 

Another method to evaluate the electrical property of particles is measuring the electrical resistance of a film pattern on a glass substrate, as shown in Figure 11b, which was fabricated with copper paste. The film pattern exhibited electrical resistivity of 4.94 Ω·cm without the sintering process, which is about 5 times higher than the graphene-wrapped Cu NPs reported earlier [14].

## 4. Conclusions

The mesosphere of carbon-shelled copper nanoparticles (Cu@MC) was prepared by direct carbonization of the organic polymer soft templates of the copper nanoparticles. The antioxidative properties of Cu@organic and Cu@MC were stable in the air up to 175 and 250 °C, respectively. The preparation method had a shortened production time and no requirement for extra carbon sources. This work demonstrates that using rich aromatic functional soft polymer templates as in situ carbon source can be an effective way to prepare Cu@MC and other carbon-shelled meso/nanoparticles. Furthermore, the produced carbon-coated copper nanoparticles and their composite paste had excellent stability and electrical conductivity. The electrical resistivity can reach 7.27 × 10^−3^ Ω·cm at room temperature, showing high potential in the field of electric sensors and devices.

## Figures and Tables

**Figure 1 materials-15-07536-f001:**
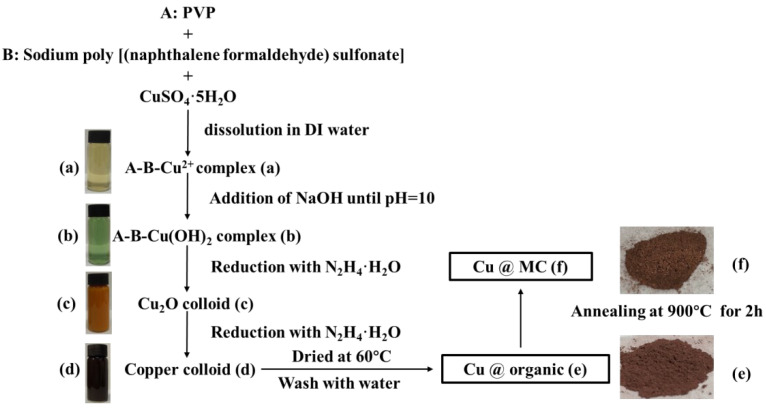
Scheme of synthesis process of Cu@MC and corresponding photographs of the samples. (**a**) A-B-Cu^2+^ complex; (**b**) A-B-Cu(OH)_2_ complex; (**c**) Cu_2_O colloid; (**d**) Copper colloid; (**e**) Cu@organic; (**f**) Cu@MC.

**Figure 2 materials-15-07536-f002:**
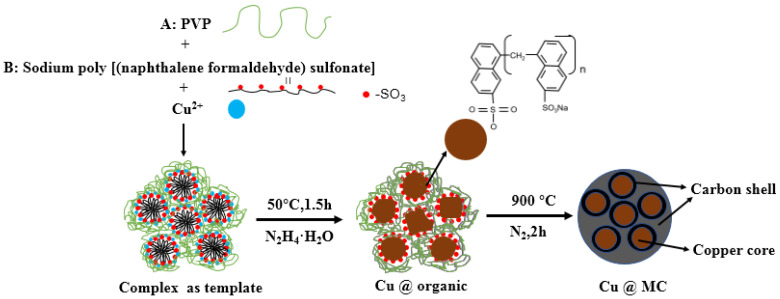
The proposed mechanism of forming Cu@MC.

**Figure 3 materials-15-07536-f003:**
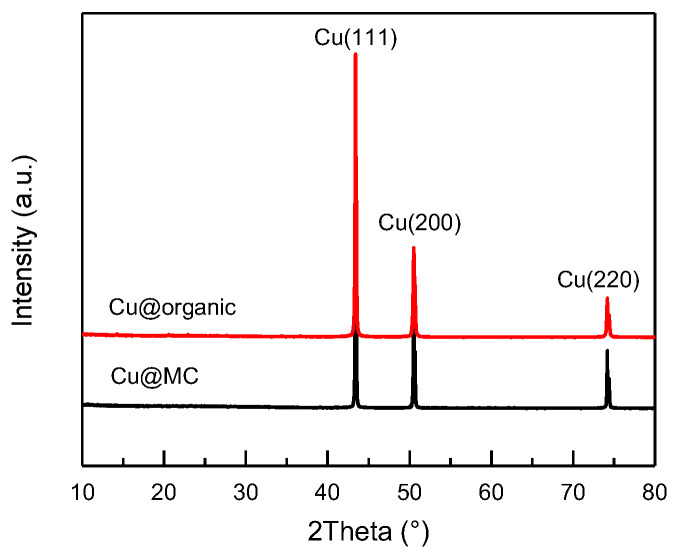
XRD patterns of Cu@organic (red) and Cu@MC (black).

**Figure 4 materials-15-07536-f004:**
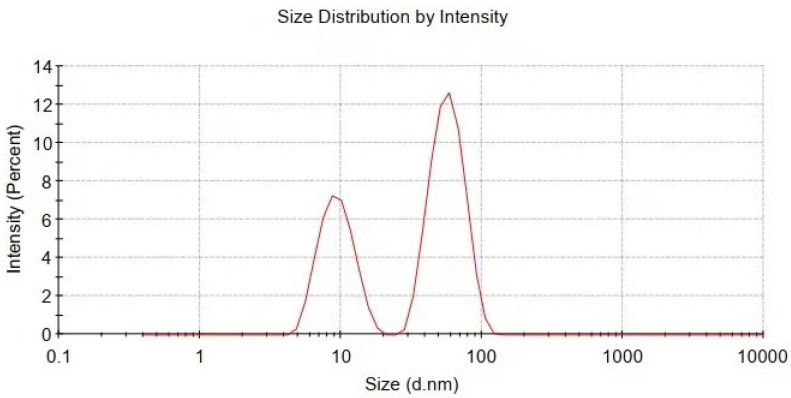
Particle size and distribution of Cu@organic characterized by dynamic light scattering measurement.

**Figure 5 materials-15-07536-f005:**
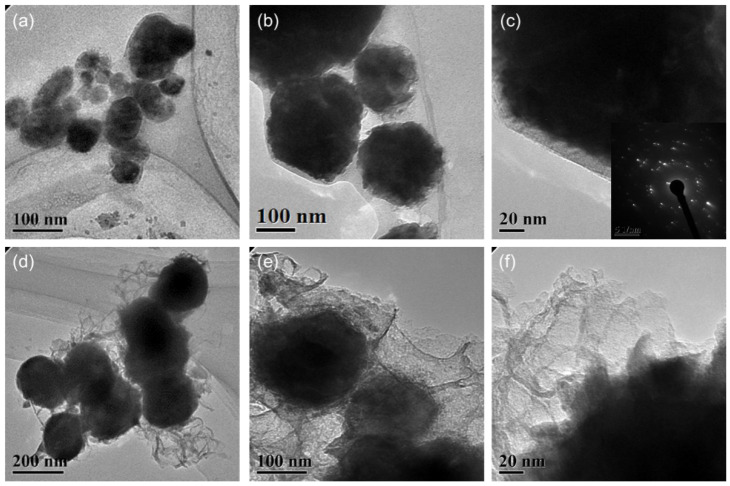
TEM images of the (**a**–**c**) Cu@organic and (**d**–**f**) Cu@MC.

**Figure 6 materials-15-07536-f006:**
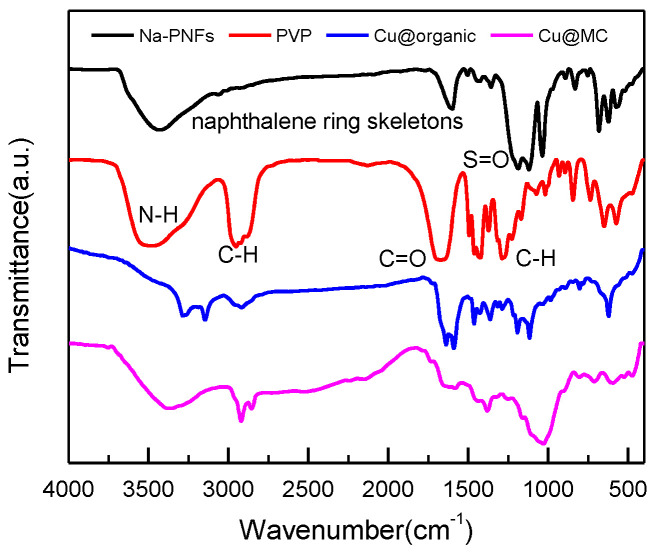
FT-IR spectra of Na-PNFS, PVP, Cu @organic, and Cu@MC.

**Figure 7 materials-15-07536-f007:**
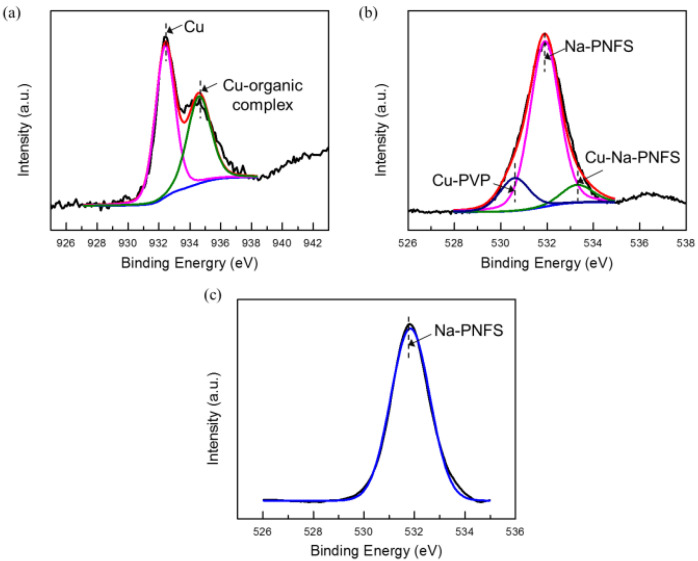
XPS spectra of (**a**) Cu 2p3/2 (raw line in red, with corresponding fitted two peaks, Cu in pink line, Cu-organic complex in green line); (**b**) O1s of Cu@organic (raw line in red, with corresponding fitted three peaks, Cu-PVP in blue line, Na-PNFs in pink line, Cu- Na-PNFs in green line); (**c**) O1s of Na-PNFS.

**Figure 8 materials-15-07536-f008:**
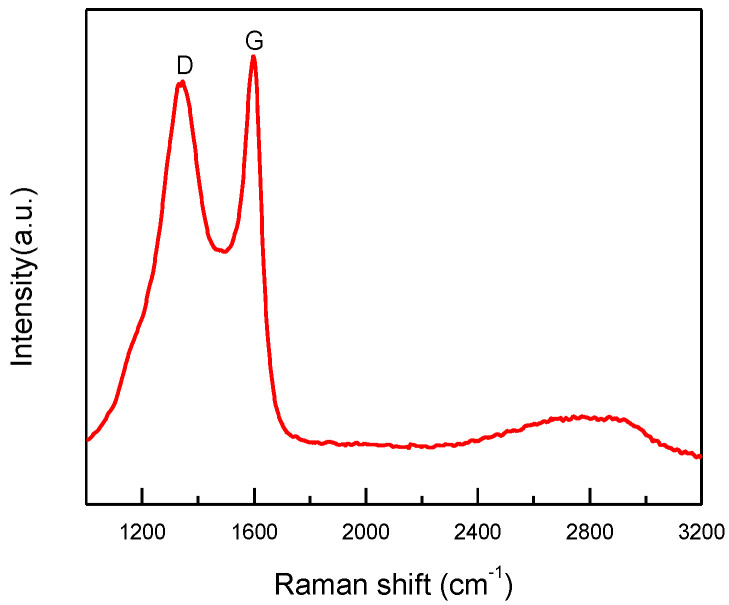
Raman spectrum of Cu@MC.

**Figure 9 materials-15-07536-f009:**
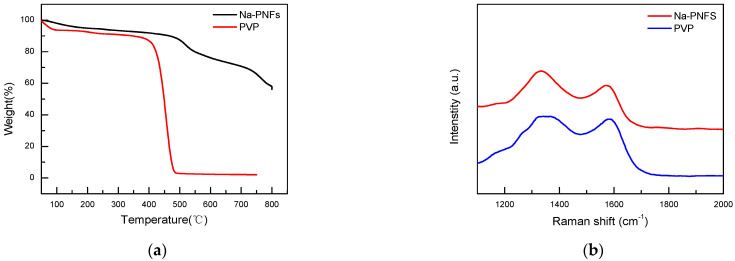
(**a**) TGA curves of Na-PNFS and PVP under nitrogen atmosphere; (**b**) Raman spectra of Na-PNFS and PVP residue after annealing at 900 °C for 2 h in nitrogen atmosphere.

**Figure 10 materials-15-07536-f010:**
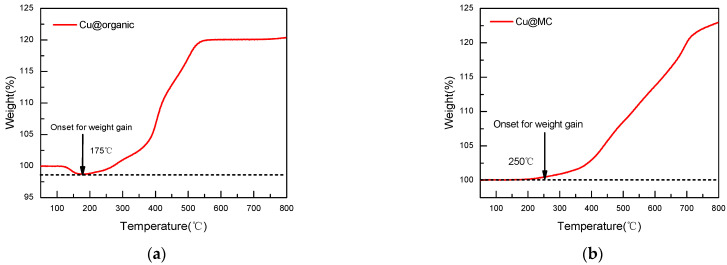
TGA curves of (**a**) Cu@organic; (**b**) Cu@MC under air atmosphere.

**Figure 11 materials-15-07536-f011:**
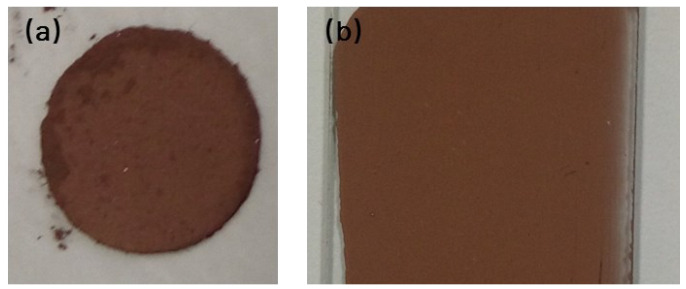
(**a**) The circle flake pressed under 20 MPa; (**b**) the film prepared by copper conductive paste on the glass substrate.

**Table 1 materials-15-07536-t001:** Electrical resistivity of the carbon-coated copper powder.

Sample	Resistance	Volume Resistivity	Pressure	Thickness	Machine
C/Cu	595 μΩ	7.27 × 10^−3^ Ω·cm	20 MPa	0.64 mm	FT-300I

## Data Availability

The data presented in this study are available from the corresponding authors upon reasonable request.

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
