# Peer review of "Mesosphere of Carbon-Shelled Copper Nanoparticles with High Conductivity and Thermal Stability via Direct Carbonization of Polymer Soft Templates"

_materials, 2022, doi:10.3390/ma15217536_

Round 1

Reviewer 1 Report

The manuscript written by Huang et al. titled ‘Mesosphere of Carbon Shelled Copper Nanoparticles with 2 High Conductivity and Thermal Stability via Direct Carboniza-3 tion of Polymer Soft Templates’ describes the syntheses of Cu on polymer which electrically conductive, thermally stable, with resistivity of 7.2×10-3 Ω•cm and can be applied in devices. 

The following points needs to be addressed 

  1. EDX analysis can be performed to understand the atomic weight percent of the Cu@organic materials 

  1. The following references maybe included 10.1021/acsomega.8b00968 ,  10.1021/acsomega.6b00447  , 10.1021/acsami.5b12714

Author Response

We sincerely appreciate you for the efforts and time put into the reviewing of the manuscript. The comments are constructive and helpful to improve the manuscript. We have carefully considered the comments and added more characterization and revised manuscript accordingly. We now submit the point-by-point reply to you. Please see the attachment.

Reviewer 2 Report

The paper describes synthesis of Cu nanoparticles claimed to be encapsulated into a carbon shell. A number of methods is used for the characterization; however, the presented results are often fragmentary, quality of some figures and included images is low, some statements are not properly grounded by the obtained data. The main conclusion about anti-oxidation ability of Cu@MC is not properly proven. Major revision is required and following comments need to be considered.

11.      Statement in line 31 “copper is relatively unstable compared to noble gas” has no sense.

22.     When the authors describe electrical conductivity, it is recommended to put it in units of Sm/cm. If they use Ohm.cm then it must be a resistivity. They should use either one or another for easier comparing the numbers and do not confusing the readers.

33.  In section 2.4, the method to obtain surface electrical resistivity is described, however, in Table 1 the volume value is presented. How the transition is made? Why the measurements were done at pressure of 20 MPa?

44. In lines 98-99, the details of the experimental phenomena are promised in the supporting materials which are not provided along with the manuscript. Therefore, the statement on “uniform stable colloidal dispersion” in lines 100-101 is not proven.

55.  Colour images in figure 1 are too small to see any details except may be a colour change.

66. It is recommended to use the same abbreviations for Cu@organic powder and Cu@MC through the manuscript. Different version can be found in figures 1, 2, 5, 6, 9 and in the text.

77. Figure 3 illustrates two main size distributions around 10 and 60-70 nm. In some panels of figure 4, one can see that bigger “particles” are actually agglomerates of smaller ones. This issue should be discussed in more detail. Also, quality (contrast) of TEM images must be improved. Some of them, especially (c), are simply black, do not allowing to see the structures. What is the meaning of the diffraction pattern presented in panel (c)?

88.  Why intensity of Cu peaks in figure 5(a) is decreased for Cu@MC?

99. The XPS spectra presented in figure 6 are not informative enough and do not support the related statements. Metallic Cu and Cu2O have nearly the same peak position in the spectra. Therefore, one can not make a conclusion about pure Cu or oxide presence. The spectrum must be taken for wider interval up to 960 eV to see Cu 2p1/2 line and possible satellites. Since, Cu can also make CuO oxide, this extension can provide grounded conclusions on the oxidation. As an example, one can refer to a publication on Cu NP oxidation with time in Applied Nano 2 (2022) 102.

110.   In line 156 it must be figure 4, not 3.

111.   The statement about “graphite-ordered carbon” in line 161 is not grounded enough. For Raman signal of graphite, one needs to address not only D/G ratio but also 2D peak intensity and width.

112.   Overall conclusion about increased stability of Cu@MC against oxidation is not proven. First, one can see from the TEM images that the shell is not continuous. Second, XPS analysis depth is about 2 nm. If the shell was continuous, one could not see signals from copper. Why XPS for Cu@organic is not compared with Cu@MC?

S   Since copper is known for gradual oxidation in ambient atmosphere, it is important to show how XPS spectra and electrical conductivity/resistivity of the obtained composites evolve with time if kept in ambient atmosphere.

Author Response

(The authors gave the same response as above.)

Reviewer 3 Report

The manuscript “Mesosphere of Carbon Shelled Copper Nanoparticles with High Conductivity and Thermal Stability via Direct Carbonization of Polymer Soft Templates” describes the preparation and characterization of mesosphere of carbon shelled copper nanoparticles.

My observations are:

 - First of all, the presentation of the results is very brief, the information obtained should be detailed and more in-depth correlations should be made. For example, the XRD spectra are presented in only one sentence, without describing the observed peaks. From the XRD spectra, the authors can calculate the size of the crystallites, to see if it changes after calcinations.

- An explanation why the diameter of the particles increases after annealing.

- The text must be unitary, the authors discuss TEM, then XRD and again TEM.

- In Fig. %b, the FTIR spectrum for Cu@MC should be added to prove that the bands characteristic to polymers disappeared from the spectrum

- The UV–vis spectra to follow specific localized surface plasmon resonance absorption of Cu NPs should be included in the manuscript

- In the literature are reported systems with Cu NPs displaying superior electrical resistivity as the values reported in this paper (Applied Surface Science 290 (2014) 240– 245, Journal of Alloys and Compounds 649 (2015) 1156e1163), and also, in reference [14] Co-based Nanoparticle@Mesoporous Carbon Nanospheres are described, not Cu NPs.

- On row 99, the authors specified something about “supporting materials”. I did not see these materials.

- Some typing/grammatical errors can be found in the manuscript.

Author Response

(The authors gave the same response as above.)

Round 2

Reviewer 2 Report

The revised version is appropriately improved and can be recommended for publication. The authors should still make a spell check. I found that in line 134, it should be Fig. 5 not 6. It could be more minor mistakes.

Author Response

Thanks for your carefully reading, and we are really sorry for the error. It should be “Fig.5”. In the revised manuscript, corresponding correction has been made. We have checked the whole content and revised the manuscript. Modifications made in the manuscript are highlighted in blue color.

Reviewer 3 Report

The authors have improved the manuscript by adding the required information, however, the typing/grammatical errors have not been corrected in the text.

Author Response

Thanks for your carefully reading and we are really sorry for the errors. We have checked the whole content and revised the manuscript. Modifications made in the manuscript are highlighted in blue color.